# Major Active Metabolite Characteristics of *Dendrobium officinale* Rice Wine Fermented by *Saccharomyces cerevisiae* and *Wickerhamomyces anomalus* Cofermentation

**DOI:** 10.3390/foods12122370

**Published:** 2023-06-14

**Authors:** Li Yao, Xueqin Shi, Hang Chen, Lin Zhang, Lanyan Cen, Lian Li, Yiyi Lv, Shuyi Qiu, Xiangyong Zeng, Chaoyang Wei

**Affiliations:** 1Key Laboratory of Plant Resource Conservation and Germplasm Innovation in Mountainous Region (Ministry of Education), Institute of Agro-Bioengineering, College of Life Sciences, Guizhou University, Guiyang 550025, China; liyao565@163.com; 2Key Laboratory of Fermentation Engineering and Biological Pharmacy of Guizhou Province, School of Liquor and Food Engineering, Guizhou University, Guiyang 550025, China; shixueqin1004@163.com (X.S.);; 3Sichuan Langjiu Co., Ltd., Luzhou 646000, China

**Keywords:** *Dendrobium officinale*, cofermentation, *Saccharomyces cerevisiae*, *Wickerhamomyces anomalus*, metabolites

## Abstract

Rice, supplemented with *Dendrobium officinale*, was subjected to cofermentation using *Saccharomyces cerevisiae* FBKL2.8022 (*Sc*) and *Wickerhamomyces anomalus* FBKL2.8023 (*Wa*). The alcohol content was determined with a biosensor, total sugars with the phenol–sulfuric acid method, reducing sugars with the DNS method, total acids and total phenols with the colorimetric method, and metabolites were analyzed using LC-MS/MS combined with multivariate statistics, while metabolic pathways were constructed using metaboAnalyst 5.0. It was found that the quality of rice wine was higher with the addition of *D. officinale*. A total of 127 major active substances, mainly phenols, flavonoids, terpenoids, alkaloids, and phenylpropanoids, were identified. Among them, 26 substances might have been mainly metabolized by the mixed-yeasts fermentation itself, and 10 substances might have originated either from *D. officinale* itself or from microbial metabolism on the newly supplemented substrate. In addition, significant differences in metabolite could be attributed to amino acid metabolic pathways, such as phenylalanine metabolism and alanine, aspartate, and glutamate metabolism. The characteristic microbial metabolism of *D. officinale* produces metabolites, which are α-dihydroartemisinin, alantolactone, neohesperidin dihydrochalcone, and occidentoside. This study showed that mixed-yeasts cofermentation and fermentation with *D. officinale* both could increase the content of active substances in rice wine and significantly improve the quality of rice wine. The results of this study provide a reference for the mixed fermentation of brewer’s yeast and non-yeast yeasts in rice wine brewing.

## 1. Introduction

*Lactobacillus* and *Saccharomyces cerevisiae* (*Sc*) are the main fermenters of wine and are responsible for starch degradation and alcohol fermentation, respectively. Moreover, there may be interesting interactions between *S. cerevisiae* and non-*S. cerevisiae* that affect wine quality, and mixed inoculations can result in higher yields than monocultures [1,2,3]. Additionally, monobacterial and mixed fermentation strains differ in growth and reproduction, metabolites, and genetic variation, and these differences highlight the characteristics of the strains and their interactions [4]. Peng et al. [5] subjected an *S. cerevisiae* strain to monobacterial and mixed fermentation with a thermotolerant strain, and the metabolic results showed that the metabolites in the wines varied greatly between fermentation methods. The rich metabolites formed during fermentation gave the rice wine its unique flavor and texture. Chloe et al. [6] used a strain of *S. cerevisiae* and *Lactobacillus thermophilus*, *Streptococcus bacillus*, and *Lactobacillus pulsus* in mono- and mixed-yeasts fermentations of wine and showed that mixed-yeasts fermentation contributed significantly to the nonvolatile metabolites in the wine. In addition, it has been found that mixed fermentations produce a greater variety and concentration of aromatic substances than pure brewer’s yeast fermentations, increasing the floral and sweet fruit flavors of wines [7].

During the fermentation of rice wine, the fermentation of grains allows the leaching of active ingredients and the accumulation of microbial metabolism, which confers a variety of functional activities to rice wine. It has been found that gavage of glutinous rice-fermented yellow wine to constipated mice can modulate microbiota-mediated intestinal ecology, regulate intestinal and serum metabolite levels, and increase gastrointestinal motility, exerting a therapeutic effect on constipation [8]. In addition, rice extracts fermented by *Aspergillus oryzae* have also been found to be a source of anti-influenza drugs and drugs for the development of anti-influenza compounds [9]. *D. officinale* is a medicinal and food plant containing a variety of bioactive components, such as polysaccharides, biphenyls, phenanthrenes, and flavonoids. Moreover, *D. officinale* is widely used as an herbal medicine with various antitumor, gastrointestinal protection, antidiabetic, immunomodulatory, and antiaging effects [10]. In addition, it has been found that fermented herbs have a better taste, as the fermentation reduces the herbaceous and bitter taste and highlights the herbs’ fruitiness. This results in beneficial changes in herbs, providing not only good effects but also good taste [11].

The current focus on mixed-yeasts fermentation with different strains is mostly on volatile metabolites, with relatively little research on nonvolatile metabolites. The physicochemical and major active metabolites, especially nonvolatile metabolites, characterizing the mixed fermentation of rice wine with the addition of *D. officinale* by *S. cerevisiae* and *W. anomalus* have not been reported. The aim was to investigate the effect of mixed-yeasts fermentation on nonvolatile metabolites in *D. officinale* rice wine to provide theoretical support for the in-depth study of the effect of microbial fermentation on the active substances in *D. officinale* and to encourage *D. officinale* to be used more widely.

## 2. Materials and Methods

### 2.1. Materials and Reagents

*S. cerevisiae* FBKL2.8022 (*Sc*), conservation number M2019406, and *W. anomalus* FBKL2.8023 (*Wa*), conservation number M2019412, were both isolated and screened from Guizhou traditional lumpy branched [12] and identified for conservation by the Key Laboratory of Fermentation Engineering and Biopharmaceuticals of Guizhou Province, Guizhou, China.

*D. officinale* was purchased from Green Spring Agricultural Development Co., Ltd. in Longanba, Pingfar Village, Yunwu Town, Guiding County, Guizhou Province. Organic glutinous rice was obtained from Houge Village, Ge Town, Ganjingzi District, Dalian City, Liaoning Province.

α-Amylase and glycosylase were from Beijing Solaibao Technology Co. Glucose specimens were from Aladdin Reagent (Shanghai, China) Co. Gallic acid, rutin, potassium bromide phenol green, forinol, and sulfosalicylic acid were provided by Shanghai Yuanye Biotechnology Co. Seventeen amino acid standards were from Hebei Guangmao Biotechnology Co. (Shijiazhuang, China). Ammonium acetate and ammonia were chromatographic grade, and the other reagents were analytically pure.

### 2.2. Rice Wine Fermentation with Mixed Yeasts

The 50 g glutinous rice was washed and soaked in 250 mL distilled water overnight and drained. The samples were sealed with a stopper with a vent valve and sterilized at 121 °C for 40 min using a sterilizer. Then, 50 g of glutinous rice was mixed with 75 mL of distilled water, and 0.28 g of glycosylase and 0.945 g of amylase were added. After heating in a water bath at 60 °C for 30 min and being brought to room temperature, 5% glutinous rice mass was added to *D. officinale* powder (see Appendix A for optimized data) and inoculated with yeast (*Sc* FBKL2.8022: *Wa* FBKL2.8023 = 10:1, amount of inoculum: 1 × 10^6^ cells/mL) then kept at 30 °C for constant-temperature fermentation and weighed every 24 h. When the weight loss was less than 0.2 g/d, the fermentation was finished. Three parallel groups were set up in each group, and the relevant components in the wine were tested and compared after fermentation. The process is shown in Appendix A (control group (SW): rice wine after cofermentation by mixed yeasts (*Sc* and *Wa*); experimental group (DOSW): rice wine with the addition of *D. officinale* cofermented by mixed yeasts (*Sc* and *Wa*); blank group (DO): *D. officinale* water extract).

### 2.3. Basic Physical and Chemical Index Determinations

Two milliliters of fermentation broth was accurately measured and centrifuged at 10,000 r/min for 5 min. The supernatant was filtered through a 0.45 μm microporous filter membrane to obtain the sample to be measured. The measurements of these indicators were performed at room temperature. For the determination of alcoholic content, the sample to be measured was diluted 400 times, and 25 μL was taken. The mass of ethanol was measured by an S-10 biosensor analyzer (Shenzhen Silman Technology Co., Ltd., Shenzhen, China). Alcoholic content = mass of ethanol/density of ethanol at 20 °C. The total sugar content was determined by the phenol–sulfuric acid method [13]. The reduced sugar content was determined by the DNS method [14]. The total acid content was determined by referring to GB/T 13662-2018 “Yellow wine”.

### 2.4. Sensory Evaluation

Reference was made to T/GZSX 017-2020 “Guizhou Rice Wine” physical and chemical indicators of fermented rice wine with modifications. A 50 mL sample was taken and poured into a 200 mL beaker and placed in bright natural light. Its color and appearance were assessed by visual inspection, followed by taste and smell evaluations to identify the aroma, taste, and style. Ten professionally trained food professionals were randomly selected to evaluate the product at room temperature from four indicators (color and appearance, aroma, taste, and style) each with 25 points out of 100, and the sensory scoring results were averaged. Water was used to mask the taste before the next sample. The evaluation criteria are shown in Appendix A.

### 2.5. Determination of Active Substance Content

Polysaccharide content determination: The fermentation broth of rice wine and anhydrous ethanol was left overnight at a ratio of 1:3 and centrifuged at 10,000 r/min for 5 min. The lower precipitate was resolubilized and freeze-dried to obtain crude polysaccharide. The polysaccharide content was determined using the phenol–sulfuric acid method [13]. Total flavonoid determination is briefly 30 μL of diluted sample (1:2) in methanol (80%), and 9 μL of sodium nitrite (5%) was added; 6 min later, 18 μL of 10% AlCl_3_ was added, and 5 min later, 60 μL of sodium hydroxide (1 M) was added. Finally, the total volume was adjusted to 300 μL with distilled water. Absorbance was measured at 510 nm. The results were obtained from the standard curves of different concentrations of catechins [15]. The total phenolic contents were determined using the Folin–Ciocalteu colorimetric method [15].

### 2.6. Nontargeted Metabolomic Assay of Rice Wine Metabolites

#### 2.6.1. Extraction of Metabolites

The samples were vortexed for 30 s and centrifuged at 4 °C for 15 min at 12,000 rpm. Fifty microliters of sample was mixed with 200 μL of extract (methanol/acetonitrile = 1:1 (*V/V*), containing isotope-labeled internal standard mixture) for 30 s, sonicated in an ice-water bath for 10 min, and centrifuged at 4 °C for 15 min at 12,000 rpm for 1 h at −40 °C. The supernatant was removed by centrifugation at 12,000 rpm for 15 min at 4 °C and filtered through a 0.22 μm microporous membrane. An equal amount of each sample was mixed to obtain QC samples and stored at −80 °C before being assayed on the machine.

#### 2.6.2. Metabolite Analysis

LC–MS/MS analyses were performed using using the Thermo Scientific™ Vanquish™ UHPLC system with a UPLC BEH Amide column (2.1 mm × 100 mm, 1.7 μm) coupled to a Q Exactive HFX mass spectrometer (Orbitrap MS, Thermo, Waltham, MA, USA). The mobile phase consisted of 25 mmol/L ammonium acetate and 25 ammonia hydroxide in water (pH = 9.75) (eluent A) and acetonitrile (eluent B). The autosampler temperature was 4 °C, and the injection volume was 2 μL. A QE HFX mass spectrometer was used for its ability to acquire MS/MS spectra in information-dependent acquisition (IDA) mode under the control of the acquisition software (Xcalibur 2.2, Thermo, Waltham, MA, USA). In this mode, the acquisition software continuously evaluates the full scan MS spectrum. The ESI source conditions were set as follows: sheath gas flow rate of 30 Arb, Aux gas flow rate of 25 Arb, capillary temperature of 350 °C, full MS resolution of 60,000, MS/MS resolution of 7500, collision energy of 10/30/60 in NCE mode, and spray voltage of 3.6 kV (positive) or −3.2 kV (negative).

All samples were detected in positive (POS) and negative (NEG) ion modes, and the identified metabolites were extracted from the raw mass spectrometry data at the first level of the chromatogram. The metabolites were identified and analyzed. The nontargeted metabolomics study of rice wine was performed by comparing DOSW with SW and DOSW with DO, combined with multivariate statistical analysis such as principal component analysis (PCA) and orthogonal partial least squares-discriminant analysis (OPLS-DA). The raw data were converted to the mzXML format using ProteoWizard and processed with an in-house program, which was developed using R and based on XCMS for peak detection, extraction, alignment, and integration. Then, an in-house MS2 database (BiotreeDB, V2.1) was applied for metabolite annotation. The cutoff for annotation was set at 0.3. Metabolic pathway construction was performed using MetaboAnalyst 5.0, https://www.metaboanalyst.ca (accessed on 12 February 2022).

### 2.7. Statistical Analysis

All trials were repeated three times, and the results were analyzed with IBM SPSS Statistics 26 and are shown as the mean ± standard deviation (SD). One-way ANOVA and independent samples *t*-test analysis were used to determine statistically significant differences at the 95% confidence level (*p* < 0.05). Graphing and data analysis were performed using Excel 2019 software and Origin 2018.

## 3. Results and Discussion

### 3.1. Sensory and Physicochemical Analyses of Rice Wine Fermented with Mixed Yeasts

In the fermentation with mixed yeasts of *Sc* and *Wa*, the fermentation cycles of SW and DOSW both ended on the 10th day of fermentation, and the trend of rice wine weight loss during this process was similar (Appendix A). As shown in Table 1, compared to SW, reducing sugars, total sugars, and pH were significantly higher in DOSW, with a slight but nonsignificant difference in alcoholic content. Furthermore, the sensory score of SW was 75.70, and that of DOSW was 79.60, indicating that the sensory score of rice wine with the addition of *D. officinale* was significantly improved. The total polysaccharide content of DOSW was 3.15 mg/mL (2.11 times as much as that of SW). The total flavonoid content of mixed-yeasts fermented rice wine SW was 50.43 mg/L, and that of DOSW was 145.78 mg/L, which was 2.89 times higher. The total phenolic content of DOSW fermented with *D. officinale* (653.44 mg/L) was increased by 216.47 mg/L compared to that of SW.

### 3.2. Metabolomic Analysis of Rice Wine

#### 3.2.1. Metabolite KEGG Pathway Enrichment Analysis

The metabolites identified in rice wine were submitted to the KEGG website for enrichment analysis of metabolic pathways, and the metabolic pathways of the five pathways with the greatest variability were evaluated by constructing bubble plots of the pathway analysis results. As shown in Table 2, the number of differential metabolites between DOSW and SW was high, and the metabolic pathways involved were complex. Moreover, the metabolites of *D. officinale* fermentation produced large differences. The pathways with significant differences are shown in Figure 1.

As shown in Figure 1, the metabolic pathway in which DOSW-SW differed most significantly in the positive ion mode was phenylalanine metabolism, belonging to amino acid metabolism. Phenylalanine metabolism is the main pathway for phenolic and flavonoid metabolism in rice wine [16]. The metabolic pathway that differed most significantly in the negative ion mode was glyoxylate and dicarboxylate metabolism, belonging to the carbohydrate metabolism metabolic pathway. The differences in metabolites between rice wine fermented with and without the addition of *D. officinale* were significant, especially in the content of active substances such as phenols and flavonoids. The differences between the two rice wines may be due to amino acid metabolism pathways, such as phenylalanine metabolism and alanine, aspartate, and glutamate metabolism, which significantly increased the content of active substances in rice wine with *D. officinale*.

#### 3.2.2. Statistical Analysis of Metabolite Classification

The metabolites in each group of rice wine were further identified and classified, and the identified substances were divided into 12 categories (amino acids, peptides, and analogues; saccharides; phenols; flavonoids; terpenoids; alkaloids; phenylpropanoids; benzene and substituted derivatives; nucleosides, nucleotides, and analogues; organic acids and derivatives; lipids and lipid-like molecules; organic compounds; others) in total. The classification statistics are shown in Figure 2A, and the proportion of each type of substance is shown in Figure 2B. Among these metabolites, in terms of species, amino acids were the most abundant, with 160 species, accounting for 25.76% of the total, and phenylpropanoids were the least abundant, with only eight species, accounting for 1.29%. In addition, phenol, flavonoid, terpenoid, alkaloid, and phenylpropanoid metabolites had higher contents and better activity in rice wine, contributing more to its quality.

### 3.3. Analysis of Major Active Metabolites

A total of 127 major active substances were identified in the three groups (SW, DO, and DOSW) of rice wine, mainly phenols, flavonoids, terpenoids, alkaloids, and phenylpropanoids. As shown in Figure 3A, naringenin, 8-acetoxypinoresinol, tetramethylpyrazine, aspalathin, mulberrin, and ginkgolide C contributed to the score in the PC1 direction, while obacunone, methyl vanillate dihydroisoalantolactone, puerarin, and (-)-epiafzelechin contributed to the score in the PC2 direction. Furthermore, dihydroisoalantolactone, puerarin, pyrophaeophorbide-a, and 2-benzylidene-1-heptanol were clustered at the positive end of the first and second principal components and contributed more. Dihydroisoalantolactone belongs to the group of terpenoids, which have been associated with protection from and treatment of chronic diseases such as heart disease or cancer [17]. Puerarin has various effects such as vasodilation, cardioprotection, neuroprotection, antioxidant, anticancer, anti-inflammation, alleviating pain, promoting bone formation, inhibiting alcohol intake, and attenuating insulin resistance [18]. This indicates that rice wine is rich in nutrients. To compare the overall differences in the active metabolites in each group of rice wine, the identified active substances were subjected to principal component analysis, and PCA plots of the relationships between the total samples in positive and negative ion modes are shown in Figure 3B. The projected score values on the plane formed by the first principal component (PC1) and the second principal component (PC2) are spatial coordinates that visualize the similarities or differences between samples. In this study, all samples in each PCA scoring plot fell within the 95% confidence ellipse. The samples in each group were well clustered, indicating that the intragroup variation of the samples was small and reproducible. DO was on the positive half-axis, and DOSW was on the negative half-axis on the first principal component, indicating that the active substances in rice wine with or without microbial fermentation were significantly different on the first principal component. Furthermore, DOSW was well differentiated from DO on the second principal component and classification, indicating that the addition of *D. officinale* fermentation had a greater effect on the active substances in rice wine.

Phenols and flavonoids have been reported to have antiaging, anticancer, and antioxidant effects [19,20,21] and to contribute more to the bioactivity of rice wine. The 127 phenols, flavonoids, terpenoids, alkaloids, and phenylpropanoids identified in the three groups of rice wine were analyzed for relative quantification, and the relative content percentage table is shown in Appendix A. The stacked bar graphs of the relative percentages of each type of active metabolite are shown in Figure 3C, and the bar graphs of the number of each type of active metabolite are shown in Figure 3D. In the SW group, there were 23 phenolic metabolites with a relative content of 69.45%, 27 flavonoid metabolites with a relative content of 11.73%, 17 alkaloid metabolites with 7.64% relative content, 28 terpenoid metabolites with 9.48% relative content, and 5 phenylpropanoid metabolites with 1.81% relative content. In the DOSW group, there were 26 phenolic metabolites with a relative content of 74.31%, 33 flavonoid metabolites with 7.56% relative content, 19 alkaloid metabolites with 8.91% relative content, 35 terpenoid metabolites with 6.77% relative content, and 8 phenylpropanoid metabolites with 2.63% relative content. In terms of the types of active substances, DOSW had the most abundant types of active substances, with 121 types of DOSW active substances, 21 more than SW. In terms of the relative content of active substances, the relative content of phenolic substances in rice wine fermented with *D. officinale* mixed yeasts increased from 69.45% to 74.31%, an increase of 4.86%. It has been shown that the concentration and composition of phenolics affect the flavor and texture of wine [22]. This is in line with the previous results of the improved taste quality of rice wine. Moreover, the relative contents of alkaloids and phenylpropanoid metabolites also increased. These results showed that *D. officinale* enriched the active ingredients of rice wine.

### 3.4. Changes in The Main Active Substances in Rice Wine

#### 3.4.1. K-Means Analysis of The Main Active Metabolites

The main active substances in rice wine were further investigated based on the differences in the main components among the three groups. To investigate the relative content trends of metabolites in different groups, the relative contents of all the differential metabolites identified in all groups according to the screening criteria were normalized by Z score. Subsequently, the results were subjected to K-means clustering analysis. Metabolites with the same trends were clustered into one cluster, and a total of six clusters were clustered.

As shown in Figure 4, 31 metabolites had the highest contents in DO (Cluster 1), the contents decreased after fermentation, and whether *D. officinale* was added had no significant effect on them. These 31 substances were mainly from *D. officinale* raw materials, and their utilization by microbial fermentation was more thorough, including 2-hydroxycinnamic acid, puerarin, phloretin, glycitein, 3-hydroxybenzoic acid, mulberrin, naringenin, and coniferin. The contents of 20 metabolites were low in DO (Cluster 2), including catechin, trans-cinnamic acid, ssinapic acid, 5,7-dihydroxy-2-(4-hydroxy-3,5-dimethoxyphenyl)-4H-chromen-4-one, limocitrin, isoginkgetin, ginkgolide J, and sakuranetin. However, they were increased after fermentation and were significantly higher in rice wine fermented with *D. officinale* than in rice wine fermented without *D. officinale*. This indicates that these 20 substances in rice wine are influenced by microbial metabolism and that microorganisms can use *D. officinale* to produce such substances. The contents of 24 metabolites, such as valerenolic acid, 3-O-p-coumaroylquinic acid, syringic acid, trans-ferulic acid, phlorin, pteroside D, pterolactam, valtrate, and isosakuranin, were low in DO (Cluster 3) and increased after fermentation, and fermentation with *D. officinale* had no significant effect on them. This indicates that these 24 substances were more likely to be produced by the microorganisms’ metabolism. The contents of 17 metabolites, including epicatechin, 3′-hydroxyresveratrol, hesperidin, daidzein, catechin, quercetin, miscanthoside, cynaroside A, and sciadonic acid, were low in DO (Cluster 6) and increased after fermentation. In addition, the contents following the addition of *D. officinale* fermentation were lower than those when *D. officinale* fermentation was not added, indicating that these 17 substances were influenced by microbial metabolism and that the addition of *D. officinale* had an inhibitory effect on the production of such substances. The contents of 11 metabolites, such as vanillic acid, neohesperidin dihydrochalcone, and occidentoside, were low in SW, high in DO (Cluster 4), and highest in DOSW. It is presumed that these substances may be mainly produced through the microbial metabolism of *D. officinale*. Eight metabolites, 4-O-caffeoylshikimic acid, scoparone, poncirin, glabrone, biorobin, and ginkgolide C, were the lowest in SW, the highest in DO (Cluster 5), and slightly decreased in DOSW. It is speculated that such substances may come from *D. officinale* and can be used by microorganisms to transform into other metabolites. These substances can be divided into four main categories, phenols, flavonoids, alkaloids, and terpenoids, as shown in Appendix A. One study found that after fermentation, total phenolic and flavonoid content generally increased, which is consistent with our findings, and they found that the increase in total phenolic and flavonoid content consequently exhibited antibacterial activities [23]. Terpenoids are very important compounds that affect the aroma of wine. It has been shown that a total of 55 terpenoids were identified in Maotai wine and that terpenoids can give a more elegant and delicate smell to Maotai wine [24].

#### 3.4.2. Cluster Analysis of The Main Active Metabolites

To discuss the changes more clearly in active substances in SW, DO, and DOSW, a HCA model was developed. The differences in the relative contents of 127 active substances in the three groups of rice wine are shown in Figure 5.

*D. officinale* is rich in phenolic and flavonoid active ingredients. Phenols and flavonoids play important roles in the biological activity of fermented wine. In this study, 28 phenols and 36 flavonoids were detected. The phenols and flavonoids with high DO contents included naringin, 2-hydroxycinnamic acid, trans-cinnamic acid, methyl vanillate, vanillic acid, 3-O-p-coumaroylquinic acid, trans-ferulic acid, 3-hydroxybenzoic acid, syringic acid, sinapic acid, quercetin, phloretin, catechin, and epicatechin. These results are similar to the findings of Zhang et al. [25], who identified 21 phenolic compounds as O-glycosylflavones, C-glycosylflavones, and phenylpropanoids, such as 1-O-caffeoyl-β-D-glucoside, isoquercitrin, kaempferol, and apigenin or other derivatives. Additionally, flavonoids such as naringin, naringin, cynaroside A, and biorobin were found in this study and have been reported to be present in *D. officinale* [10,25,26]. In addition, studies have found that *D. officinale* is rich in dihydroflavonols (dihydrokaempferol and dihydroquercetin) [26]. Dihydrocapsaicin, α-dihydroartemisinin, and neohesperidin dihydrochalcone were obtained in this study. However, they have not been reported in the active ingredients of *D. officinale*, and whether these substances are from *D. officinale* needs further study.

When *D. officinale* was added and mixed yeasts (*Sc* and *Wa*) were used to ferment *D. officinale* rice wine, the phenols and flavonoids with high contents in DOSW were trans-ferulic acid, trans-cinnamic acid, 2-hydroxycinnamic acid, vanillic acid, 3-O-p-coumaroylquinic acid, methyl vanillate, syringic acid, catechin, sinapic acid, eriodictyol, naringin, and 3-hydroxybenzoic acid. Among them, trans-ferulic acid, trans-cinnamic acid, 2-hydroxycinnamic acid, vanillic acid, 3-O-p-coumaroylquinic acid, methyl vanillate, syringic acid, catechin, eriodictyol, and sinapic acid were also detected in SW (rice wine without *D. officinale* addition), and there was a difference in the content. This suggests that the high contents of phenols and flavonoids in *D. officinale* rice wine are partly from *D. officinale*, while *Sc* and *Wa* can also produce such substances through their metabolism. In this study, astragaloside 3′-hydroxy resveratrol was also detected. Astragalosides have a wide range of biological activities. Moreover, resveratrol is an important active component of *D. officinale* with anticancer activities [27,28].

### 3.5. Screening of Major Active Differential Metabolites

The metabolites with large differences in DO and DOSW were screened by variable importance in the projection (VIP) and *p* value. The active differential metabolites in the two groups of rice wine were considered differential metabolites with VIP values > 1 and *p* values < 0.05. Among the 127 active substances in DO and DOSW, 95 differential metabolites were screened, and 66 metabolites had significantly upregulated content after fermentation, while 29 metabolites had significantly downregulated content. Ninety-five differential metabolites were visualized in a Z score plot (Figure 6A) for each differential metabolite between the two groups of samples.

Figure 6B shows an UpSet plot of the 95 differential metabolites, from which the distribution of substances among groups can be visualized. As shown in the UpSet plot of metabolites and the Z score plot of differential metabolites, 51 metabolites were present in the three groups of samples at the same time. Among them, the contents of 33 substances, such as 3-(3,4,5-trimethoxyphenyl) propanoic acid, 4-hydroxy-2,6-dimethylaniline, 3-O-p-coumaroylquinic acid, α-bixin, epicatechin, daidzein, syringic acidic acid, catechin, dihydrocapsaicin, trans-ferulic acid, trans-cinnamic acid, sinapic acid, 5,7-dihydroxy-2-(4-hydroxy-3,5-dimethoxyphenyl)-4H-chromen-4-one, limocitrin, vanillic acid, phlorin, and 3′-hydroxy resveratrol, increased significantly after the fermentation of *D. officinale*. This is probably because such substances could be produced through microbial metabolism. At the same time, *D. officinale* contained such substances, and microbial fermentation promoted the dissolution of such substances. The contents of 18 substances, such as 2-hydroxycinnamic acid, phloretin, hyperforin, biorobin, curcumin III, and naringin, decreased significantly after the fermentation of *D. officinale*, and the microbial metabolism may have used these substances in *D. officinale* for metabolism and biotransformation, resulting in decreases in the contents of these substances.

Twenty-six substances were present only in SW and DOSW, mainly astaxanthin, valerenolic acid, 4-nitrophenol, phenol, pyrroloquinoline quinone, hesperidin, lucidenic acid D1, and dihydrofukinolide, which were produced by the microorganisms’ own metabolism and were significantly increased after fermentation.

Ten substances were present only in DO and DOSW, indicating that such substances were only derived from *D. officinale* raw materials or the metabolism of *D. officinale* raw materials by microorganisms. Eight substances (8-acetoxypinoresinol, β-santalic acid, tetramethylpyrazine, curcumin dimer 1, pyrophaeophorbide a, mulberrin, silidianin, and ginkgolide B) showed downward adjustments in content after fermentation, indicating that microbial fermentation utilized such substances in *D. officinale*. Two substances, geniposidic acid and demethyloleuropein, showed upward adjustments in content after fermentation. The contents of these two substances increased, possibly because microbial fermentation used other substances in *D. officinale* for biotransformation.

2-Benzylidene-1-heptanol, dihydroisoalantolactone, puerarin, and coniferin were only present in DO. Microbial fermentation utilized these four substances more thoroughly and was not detected in the fermented *D. officinale* rice wine. Furthermore, α-dihydroartemisinin, alantolactone, neohesperidin dihydrochalcone, and occidentoside were detected only in DOSW, and these four substances might be characteristic metabolites produced by the microbial metabolism of *D. officinale*.

### 3.6. Pathway Analysis of Key Active Differential Metabolites

The 95 differential metabolites in DO and DOSW were mapped through authoritative metabolite databases such as KEGG and PubChem to analyze the metabolic pathways with the highest correlations to the differential metabolites, and the results are shown in Figure 7. The pathway analysis showed that the active differential metabolites were mainly involved in five metabolic pathways, namely phenylalanine metabolism; glycine, serine, and threonine metabolism; limonene and pinene degradation; tyrosine metabolism; and arginine and proline metabolism, in *D. officinale* rice wine before and after mixed-yeasts fermentation. Among them, phenylalanine metabolism and tyrosine metabolism are the main metabolic pathways of phenolic flavonoids. The metabolic pathways of the major phenolic flavonoids and other metabolites are shown in Figure 8.

The mixed-yeasts fermentation promoted increases in the contents of metabolites such as trans-cinnamic acid, trans-ferulic acid, sinapic acid, sakuranetin, myricetin, and catechin in the rice wine. Quercetin, naringin, eriodictyol, kaempferol, catechin, epicatechin, and other substances are synthesized or converted mainly through the phenylalanine metabolic pathway. In this study, L-phenylephrine was catalyzed by related enzymes to generate trans-cinnamic acid, trans-ferulic acid, and sinapic acid. Finally, secoisolariciresinol, (-)-matairesinol, coniferin, and scoparone were synthesized under the effect of some key enzymes. Meanwhile, through the phenyl propane metabolic pathway, the related enzymes catalyzed L-phenylephrine and L-tyrosine into p-coumaroyl coenzyme A, the starting substrate for flavonoid metabolism. There are two branches for the subsequent reaction of p-coumaroyl coenzyme A. One is the formation of caffeoyl coenzyme A and feruloyl coenzyme A catalyzed by the related enzymes. The other is the formation of naringenin chalcone catalyzed by chalcone synthase; then, naringin, other flavonoids, dihydroflavonols, and flavonols are finally generated [29,30,31]. In this study, L-tyrosine in the presence of p-coumaroyl coenzyme A produced 3-O-p-coumaroylquinic acid and 4-O-caffeoylquinic acid, which were metabolized in the presence of caffeoyl coenzyme A and feruloyl coenzyme A to coniferin, (-)-matairesinol, etc. L-tyrosine in the presence of p-coumaroyl coenzyme A also produced naringin and phloretin. Phloretin could be further metabolized to phlorin, and naringin was then metabolized to hesperidin, sakuranetin, eriodictyol, kaempferol, etc., and finally metabolized to neohesperidin dihydrochalcone, isosakuranin, cynaroside A, biorobin, (-)-epiafzelechin, myricetin, catechin, and epicatechin. The microbial mixed-yeasts fermentation process reduced the contents of naringin, phloretin, and 2-hydroxycinnamic acid. The leucocyanidin in *D. officinale* rice wine could be decomposed into metabolites such as catechin and epicatechin during the fermentation process, which were ultimately retained, and the content was significantly increased in the fermented *D. officinale* rice wine.

## 4. Conclusions

Compared with SW, the DOSW sensory score, reducing sugars, total sugars, and pH were significantly higher, and the alcoholic content was slightly but not significantly different. In addition, a total of 127 major active substances, mainly phenols, flavonoids, terpenoids, alkaloids, and phenylpropanoids, were identified in the rice wine of the three groups SW, DO, and DOSW. The active substances identified were subjected to principal component analysis, and the results showed that the addition of *D. officinale* fermentation had a significant effect on the active substances in the rice wine. In particular, the contents of phenols, flavonoids, and other active substances increased significantly in the DOSW group. These differences may be caused by amino acid metabolic pathways. Additionally, in terms of the types of active substances, DOSW had the most abundant types of active substances, and in terms of the relative contents of active substances. This indicates that the fermentation of mixed yeasts was more favorable to the solubilization of phenolic substances in *D. officinale*, and the contents of some metabolites were reduced. At the same time, the contents of catechin and epicatechin increased significantly. Among 95 differential metabolites in the three groups, 26 substances were only present in SW and DOSW (rice wine fermented with mixed yeasts), indicating that they may be mainly produced by microorganism metabolism. However, it could not be determined whether these substances were metabolized by *S. cerevisiae* or *W. anomalus*. Only 10 substances were present in the samples containing *D. officinale* (DO and DOSW), indicating that they may be derived from *D. officinale* or the microbial metabolism of *D. officinale*. Moreover, α-dihydroartemisinin, alantolactone, neohesperidin dihydrochalcone, and occidentoside were detected only in DOSW, indicating that these four substances may be characteristic metabolites produced by the microbial metabolism of *D. officinale*. In addition, 95 differential metabolites were mainly enriched in five metabolic pathways (phenylalanine metabolism; nicotinate and nicotinamide metabolism; alanine, aspartate, and glutamate metabolism; glyoxylate and dicarboxylate metabolism; and riboflavin metabolism). The results of this study showed that the addition of *D. officinale* for fermentation could improve the quality of rice wine, and the mixed-yeasts cofermentation had good effects on the non-volatile metabolites in rice wine as well as the active substances in *D. officinale*. This study can provide a theoretical basis for the mixed cofermentation of brewer’s yeast and non-yeast in rice wine brewing.

## Figures and Tables

**Figure 1 foods-12-02370-f001:**
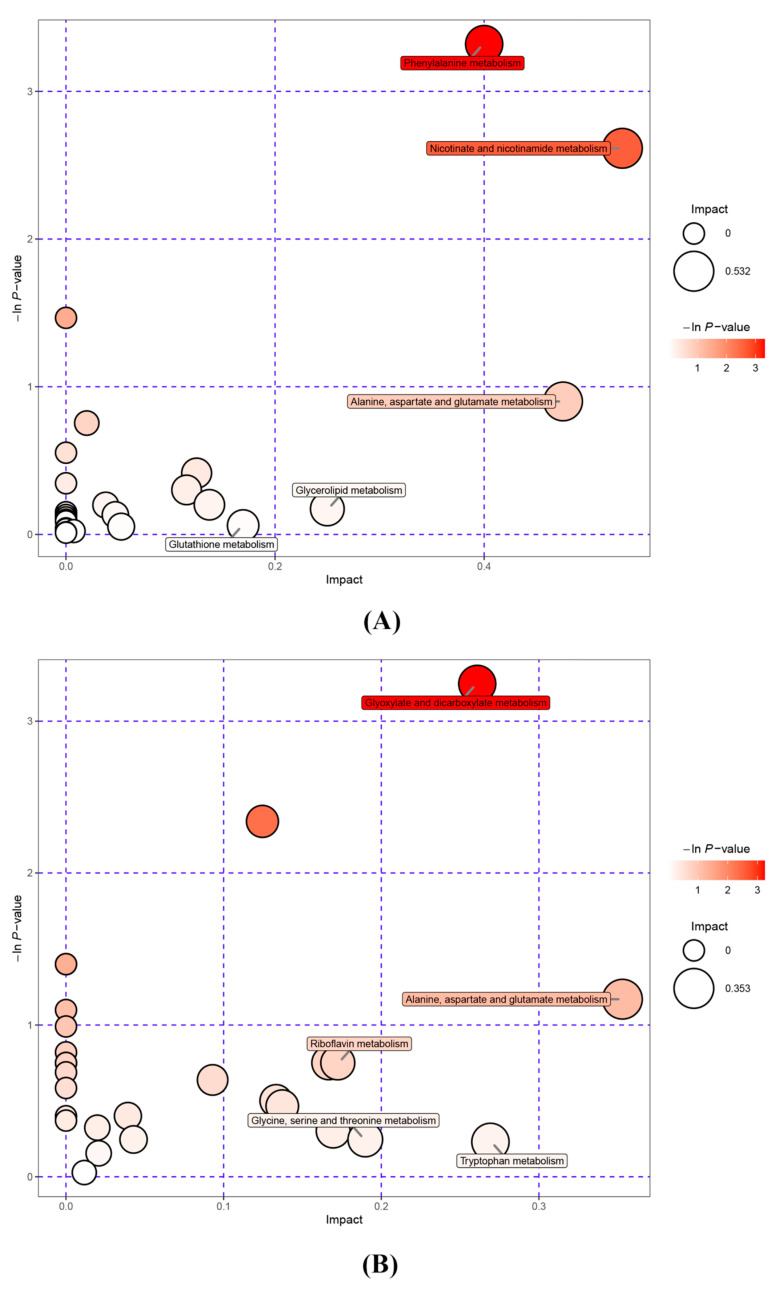
Pathway analysis for rice wine. (**A**) Pathway analysis diagram of the positive ion mode for DOSW-SW. (**B**) Pathway analysis diagram of the negative ion mode for DOSW-SW. Each bubble in the bubble diagram represents a metabolic pathway, and the horizontal coordinate and size of each bubble indicate the influence factor of the pathway in the topological analysis. The larger the bubble is, the larger the influence factor. The vertical coordinate and color of the bubble indicate the *p* value of the enrichment analysis (taking the negative natural logarithm, i.e., −lnp); the darker the color, the smaller the *p* value, and the more significant the enrichment.

**Figure 2 foods-12-02370-f002:**
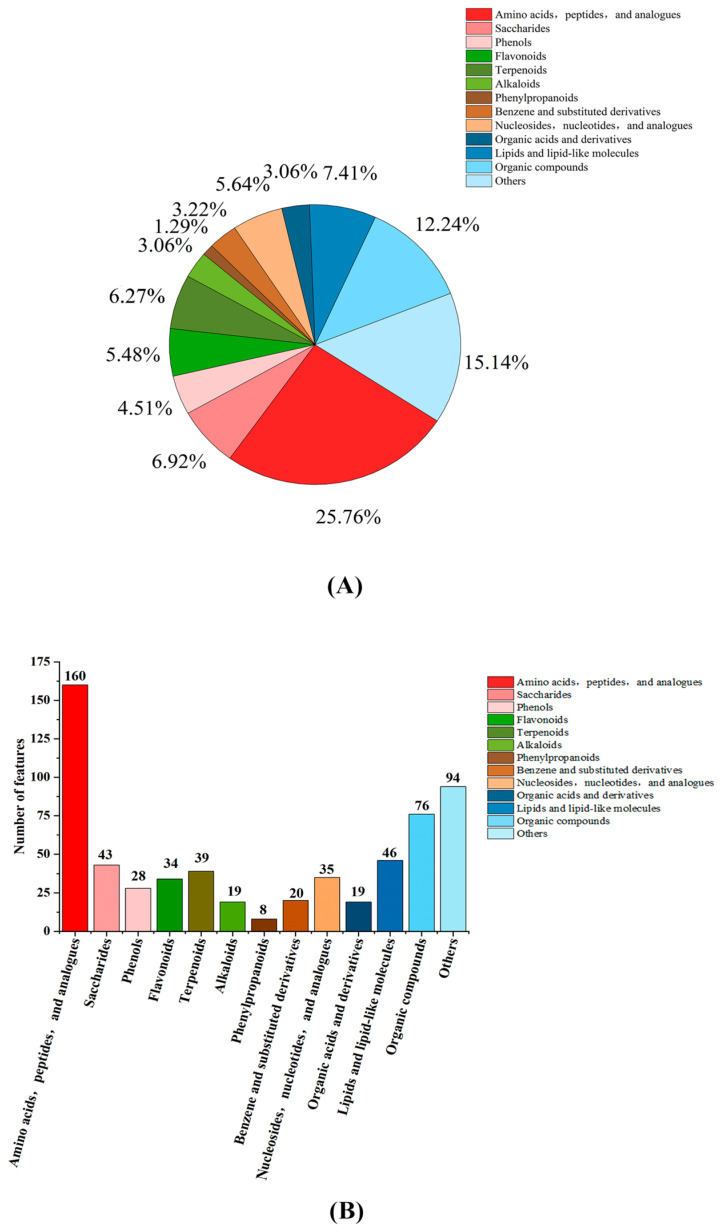
Metabolism classification statistics. (**A**) The metabolism classification statistic. (**B**) The proportion of each type of substance.

**Figure 3 foods-12-02370-f003:**
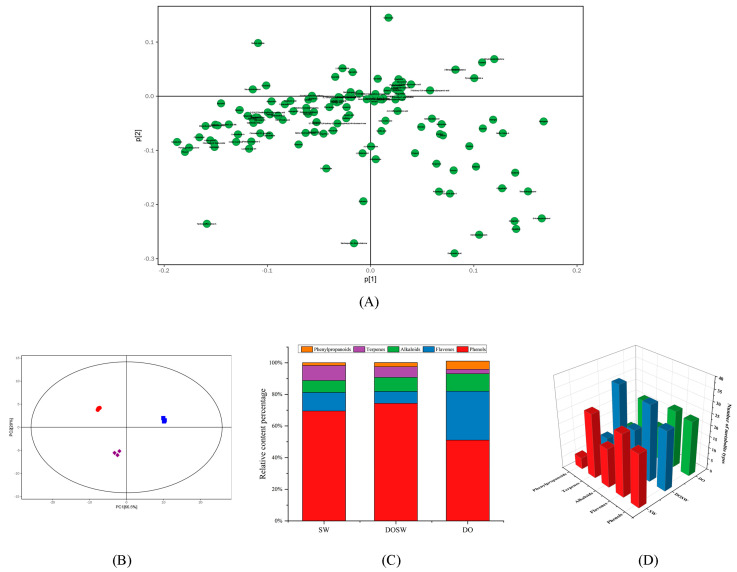
(**A**) Principal component analysis loading label plot of metabolites. (**B**) Principal component analysis score plot of metabolites. (**C**) 3D histogram showing stacked percentages. (**D**) 3D histogram showing number of metabolites.

**Figure 4 foods-12-02370-f004:**
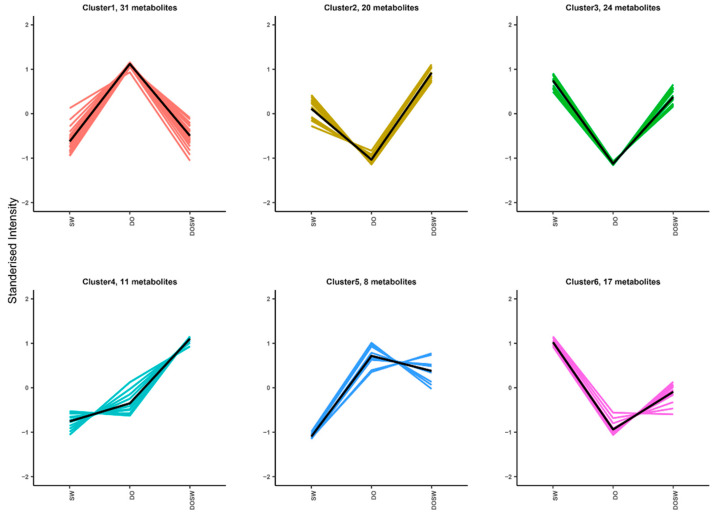
K-means plot of metabolites. The vertical coordinate indicates the relative content of the metabolite after normalization. A relative content of less than 0 means that the relative content of the substance in this group is lower than the average of the relative content in the three groups.

**Figure 5 foods-12-02370-f005:**
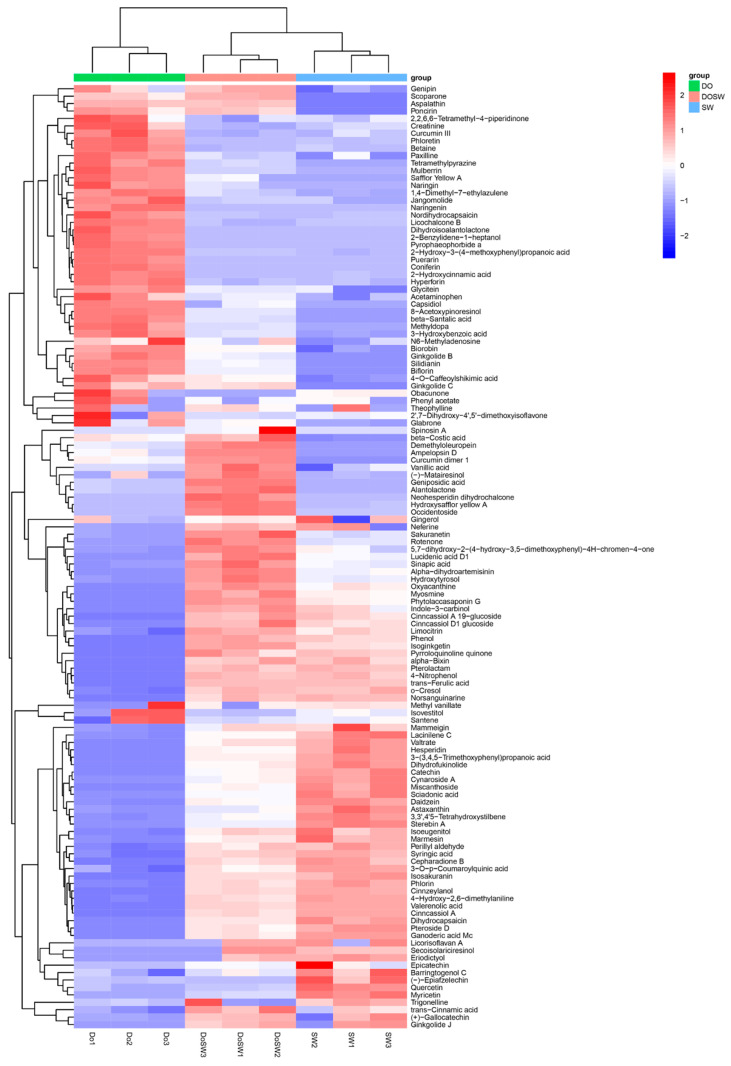
Heatmap of hierarchical clustering of metabolites. The horizontal and vertical coordinates represent the sample name and the different metabolites, respectively. From blue to red indicates metabolite expression abundance from low to high.

**Figure 6 foods-12-02370-f006:**
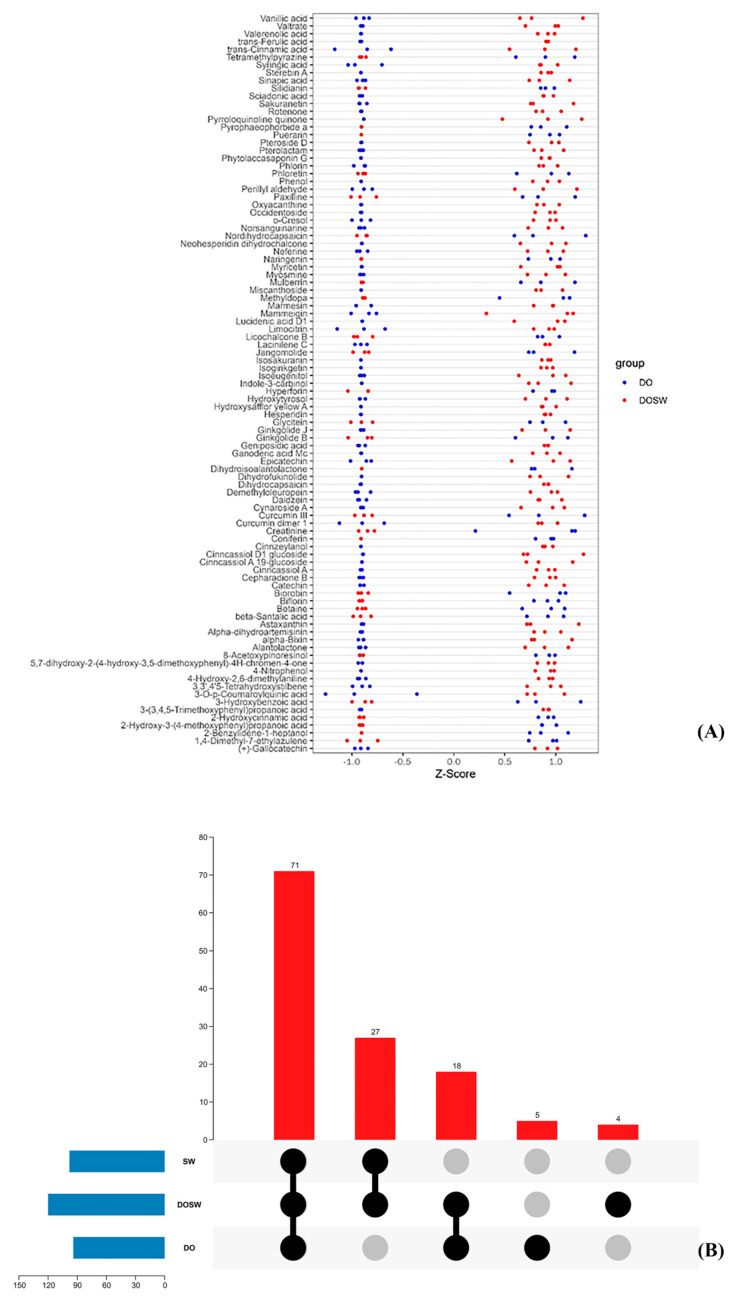
(**A**) Z score plots of differentially expressed metabolites. (**B**) UpSet plot of metabolites. The Z score values were calculated to normalize the differential metabolites of the different samples. The horizontal coordinate indicates the Z value, the vertical coordinate indicates the differential metabolites, dots of different colors indicate different groups of samples, Z < 0 means the relative content of the substance is downregulated, and Z > 0 means the content of the substance is upregulated.

**Figure 7 foods-12-02370-f007:**
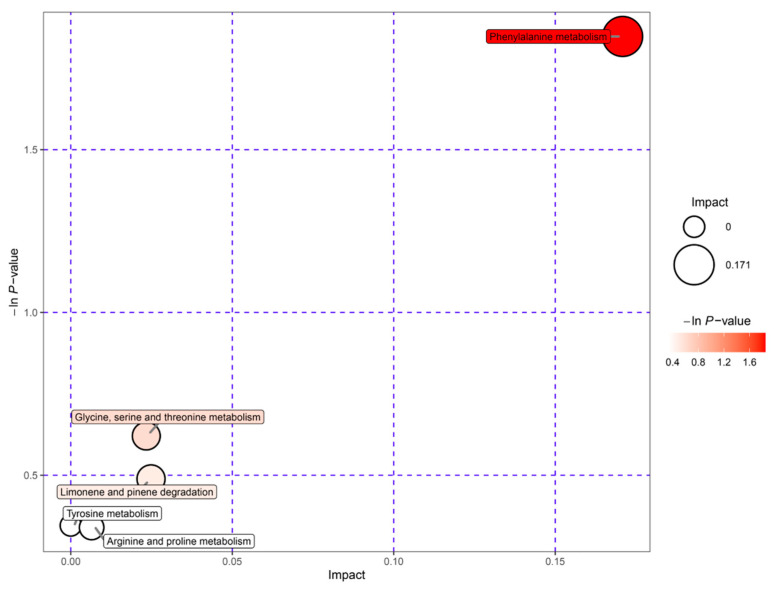
Metabolic pathway analysis of rice wine before and after fermentation. Each bubble in the diagram represents a metabolic pathway; the horizontal coordinate where the bubble is located and the bubble size indicate the influence factor size of the pathway in the topological analysis. The larger the bubble is, the larger the influence factor. The vertical coordinate where the bubble is located and the bubble color indicate the *p* value of the enrichment analysis (taking the negative natural logarithm, i.e., −lnp). The darker the color, the smaller the *p* value, and the more significant the enrichment degree.

**Figure 8 foods-12-02370-f008:**
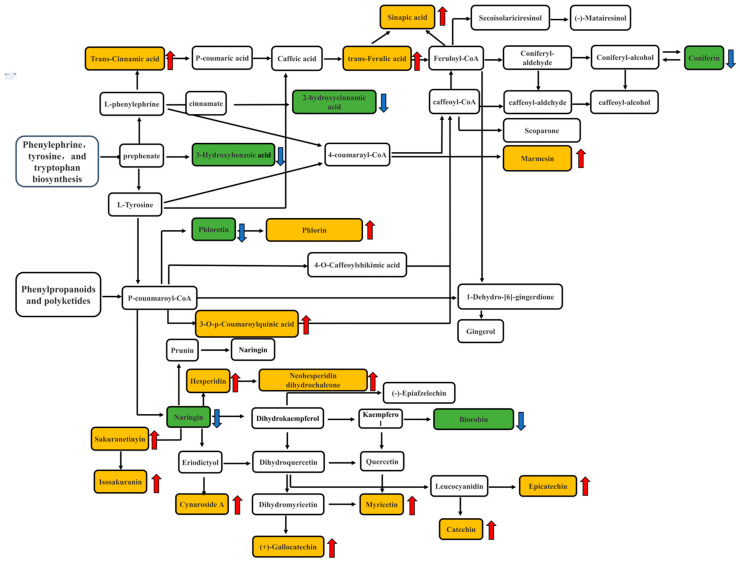
Analysis of metabolic pathways of active substances before and after fermentation of rice wine. An upward arrow indicates the upward adjustment of content, and a downward arrow indicates the downward adjustment of content. Yellow boxes are for upregulated substances, green boxes are for downregulated substances.

**Table 1 foods-12-02370-t001:** Basic physicochemical indices and sensory scores of *D. officinale* rice wine.

Fermentation Method	Alcoholic Strength (%vol)	Reduced Sugar Content (mg/mL)	Total Sugar Content (mg/mL)	Total Acid Content (mg/mL)	pH	Sensory Scoring
SW	15.10 ± 0.42 ^a^	16.25 ± 0.38 ^b^	42.88 ± 1.02 ^b^	3.02 ± 0.10 ^a^	4.65 ± 0.06 ^b^	75.70 ± 3.23 ^b^
DOSW	14.60 ± 0.74 ^a^	23.71 ± 0.14 ^a^	61.81 ± 3.06 ^a^	3.02 ± 0.05 ^a^	4.95 ± 0.06 ^a^	79.60 ± 3.98 ^a^

Notes. Different lowercase letters indicate significant differences. SW: Rice wine after cofermented by mixed yeasts (*Sc* and *Wa*). DOSW: Rice wine with the addition of *D. officinale* cofermented by mixed yeasts (*Sc* and *Wa*).

**Table 2 foods-12-02370-t002:** Main enrichment results of the KEGG pathway of metabolites.

	DOSW-SW		DOSW-DO	
Ion mode	ES+	ES−	ES+	ES−
Metabolites	516	479	524	520
Differential metabolites	192	183	327	399
Up-metabolites	166	169	314	399
up-Metabolic pathway	50	48	57	66
Down-metabolites	26	14	13	0
Down-metabolic pathway	15	7	9	0

## Data Availability

The data presented in this study are available on request from the corresponding author.

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
