# Peer review of "Major Active Metabolite Characteristics of *Dendrobium officinale* Rice Wine Fermented by *Saccharomyces cerevisiae* and *Wickerhamomyces anomalus* Cofermentation"

_foods, 2023, doi:10.3390/foods12122370_

Round 1

Reviewer 1 Report

Dear Authors, please see the detail in the file attached where some parts of the text in the manuscript were highlighted and commented by me. The amount of the imperfections is quite significant and must be corrected

I think that the whole manuscript should be checked both about the English, but also from the "clearness" point of view

Reviewer 2 Report

This an interesting article about supplementing rice wine with Dendrobium officinale during cofermentation and analysis the wine using basic physiochemical parameters and metabolomics. The results are interesting and contribute to the current knowledge. However, the authors must work to improve the content before reconsideration for publication. Also some figures are blur and difficult to read …please add high quality figures. The grammar needs improvement as well. Below comments could help improve the manuscript  

Line 16 Mixed not mixes

Line 15-16 Dendrobium officinale rice wine was cofermented with Saccharomyces cerevisiae FBKL2.8022 (Sc) and Wickerhamomyces anomalus FBKL2.8023 (Wa) ???? this reads better. Please consider adapting.  

Line 16-17 “Three groups were used to investigate the physicochemical and main active metabolite characteristics” doesn’t make sense please…what group are you talking about.

Abstract should include brief description of the subject, methodology (which is lacking in this abstract), and promising results. Please include methodology.

Line 18 what do you mean by “quality” in this context

Line 20 “in these three groups” again this doesn’t add up. Or if you add the methodology and explain what the 3 groups means then it should be fine.

Line 22  metabolites ………… delete metabolite contents

Line 24  please delete “Additionally” Also rephrase line 24 -26

Line 26  mixed yeasts fermentation and fermentation should be “cofermentation”

Line 35 delete are

Line 42  followed the metabolic results ….please rephrase

Line 70-74 move to methods. This shouldn’t be here

Line 74 to 76 please delete

Line 99 reference (Figure S1) in the last paragraph of that section. Also delete “The process is shown in Figure S1”

Line 98 please improve this section. Your descriptions are difficult to follow and doesn’t align well with the FS1.

Line 100 what was used for the sterilization? Please add

Line 101 1.5 times???

Your FS1 please check and improve it. What is expand training (do you want to say yeast activation?). Also why do you add yeast before water bathing at 60 degrees? Wont it kill the yeast? Check how you reported the yeast numbers 6 should be superscripted

Line 109 Rice wine after fermentation by mixed yeasts (Sc and Wa)….should be Rice wine co-fermented with Sc and Wa????

Line 112 poorly descried. Werent the panalist trained, what volume was served, what was used mask taste before next samples, room temperature etc. Please improved

Line 118 should come before line 112. Please rearrange your methodology to flow well. Also poorly described. Improve please

Line 128 please improve with details

Why was volatile organic compounds (esters, etc) measured? This could have add value to the manuscript. Please consider adding this 

The authors should improve the grammar 

Reviewer 3 Report

In my opinion, the manuscript entitled Major Active Metabolite Characteristics of Dendrobium officinale Rice Wine Fermented by Saccharomyces cerevisiae and Wickerhamomyces anomalus Cofermentation by Yao et al., is a very good article. The introduction is well written, the materials and methods are enough described and the results are explained and compared with the current state of the art.

I have some comments as follows:

1. line 101-102 – please rephrased it. For me it is not clear which was the exact addition of glycosylase, amylase and distilled water and what does it means 1.5 times?

2. line 106 which is the abbreviation used for 0.6 g/d – d?

3. line 116: what does it mean the indicator style? It is used for describing which organoleptic attributes?

4. line 121: The should be written with the

5.in the conclusion, authors should present the obtained results and not to give explanations. For instance lines  488-489. 

Round 2

Reviewer 2 Report

The manuscript has been improved and i recommend accepting in the present form

It reads well

Author Response

Thank you!